# MAC-TIGER: MULTI-AGENT COOPERATION FOR ENHANCED TEXT-TO-IMAGE GENERATION

## ABSTRACT

Recent advancements in text-to-image (T2I) generation have improved image fidelity and alignment with textual prompts, but challenges remain in handling complex compositional tasks, such as attribute binding, spatial relationships, and numerical precision. To address these issues, we propose Mac-Tiger, a novel multi-agent cooperation framework that leverages multimodal large language models (MLLMs) to optimize T2I generation through iterative refinement. Unlike conventional methods, Mac-Tiger employs a tri-agent system—comprising Reviewer, Challenger, and Refiner roles—that collaboratively evaluates and refines prompts through dynamic feedback loops and multimodal analysis. In the first phase, the agents generate initial solutions by exploring diverse perspectives and evaluating each other's outputs. In the second phase, the agents iteratively refine the prompt by addressing gaps and inconsistencies identified during the review and challenge phases. This cooperative feedback process results in higher-quality outputs and more robust, context-aware image generation. Experiments on benchmarks such as T2I-CompBench and MagicBrush demonstrate that Mac-Tiger outperforms state-of-the-art methods, achieving higher quality and more context-aware image generation. Our code is available [1].

## 1 INTRODUCTION

Recent advancements in diffusion models Ho et al. (2020); Dhariwal & Nichol (2021) have significantly enhanced the quality and aesthetic appeal of generated images, making text-to-image (T2I) generation a key area of AI research. These improvements can be classified into two main approaches: one focuses on fine-tuning pre-trained models for specific tasks, resulting in strong performance for specialized applications Li et al. (2024); Chen et al. (2023b); Ruiz et al. (2023), while the other builds models from scratch to improve generalizability and overall image quality Ramesh et al. (2022); Rombach et al. (2022); Podell et al. (2023); Chen et al. (2023c).

However, despite these advancements, challenges remain when generating images from complex prompts. Models often struggle to cohesively assemble multiple objects, spatial relationships, or intricate attributes into a single, consistent image. As demonstrated in Figure 1, while methods like SDXL and DALL-E 3 produce impressive results, they sometimes fail to handle complex interactions or precise spatial placements. Furthermore, the inherent randomness in diffusion models can lead to outputs that deviate from or contradict the user's prompt. In addition, vague or brief prompts further complicate the generation process, as these models lack effective strategies for refining them, resulting in inaccuracies in the generated images.

To address these challenges, previous work has focused on single-agent methods for optimizing prompts through iterative refinement Hao et al. (2024); Brade et al. (2023). However, these approaches treat the process as a single-agent task and are limited by the model's inability to evaluate and correct errors, especially in complex scenarios. Multi-agent cooperation Hong et al. (2023); Ferber & Drogoul (1992); Liu et al. (2024) offers a promising alternative, where multiple agents collaborate to refine solutions through feedback loops. While multi-agent systems have been explored in other domains, their potential in T2I generation remains largely unexplored, leaving a significant gap in the field.

---

[1]https://anonymous.4open.science/r/Mac-Tiger-6635/

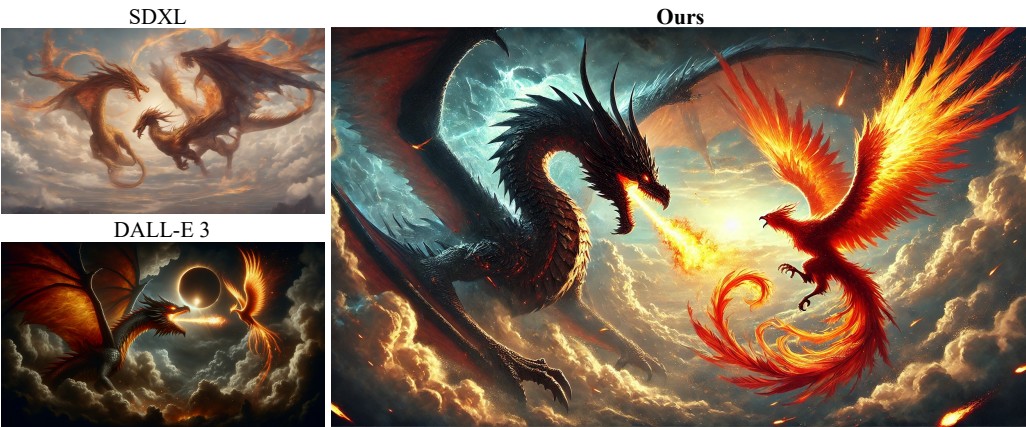

**Prompt:** In the vast sky, a fierce battle unfolds. On the left, **a dragon** with **shimmering scales** and **blazing eyes** breathes scorching fire, its **tail coiling** destructively. On the right, **a phoenix** with **fiery wings** and **radiant feathers** emits sparks, its eyes gleaming with wisdom. They confront each other, circling and weaving through clouds. Each clash **sparks lightning**, illuminating the sky.

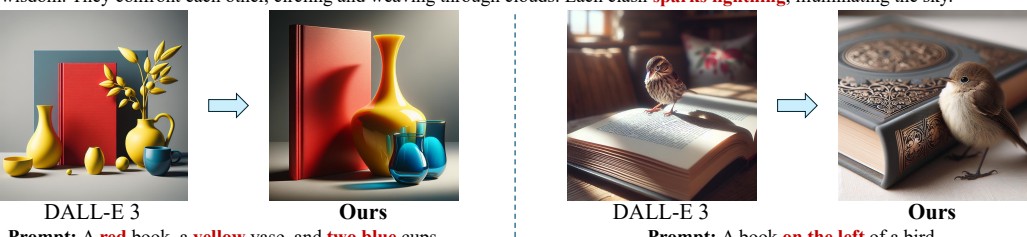

**Prompt:** A **red** book, a **yellow** vase, and **two blue** cups.                    **Prompt:** A book **on the left** of a bird

Figure 1: Comparison of Mac-Tiger with SDXL and DALL-E 3 on text-to-image generation tasks. Mac-Tiger, utilizing multi-agent cooperation, demonstrates superior ability in accurately capturing intricate details and compositional elements in complex prompts, such as the interaction between a dragon and a phoenix or the precise placement of specific objects.

In this work, we propose Mac-Tiger, a novel multi-agent cooperation framework designed to address the limitations of single-agent approaches in text-to-image generation. Unlike traditional methods, which typically rely on a single model to optimize prompts and generate images, Mac-Tiger employs a tri-agent system consisting of a *Reviewer*, *Challenger*, and *Refiner*. These agents work collaboratively to iteratively evaluate and refine the prompt, overcoming issues such as inconsistencies, vague descriptions, and complex compositional tasks. In the first phase, the *Reviewer* assesses the initial prompt, identifying potential gaps and inconsistencies. The *Challenger* then critiques the reviewer's evaluation, identifying overlooked issues or contradictions, and proposing new constraints or perspectives. Finally, the *Refiner* synthesizes feedback from both the *Reviewer* and *Challenger*, generating an optimized prompt for image generation. This iterative feedback loop continues until a high-quality, contextually accurate prompt is produced.

Through this multi-agent feedback system, Mac-Tiger effectively improves the accuracy, coherence, and overall quality of the generated images. Unlike single-agent methods that may be limited by a narrow perspective or unable to adapt to complex prompts, our framework benefits from diverse viewpoints and dynamic problem-solving, enabling more robust and reliable T2I generation. Experiments on benchmark datasets such as T2I-CompBench and MagicBrush demonstrate that Mac-Tiger outperforms state-of-the-art approaches in both task completion rate and image quality.

## 2 RELATED WORK

**Image generation and editing.** Diffusion models Ho et al. (2020); Dhariwal & Nichol (2021) have driven advances in image generation/editing, with high-quality T2I frameworks Rombach et al. (2022); Saharia et al. (2022); Chen et al. (2023c); Podell et al. (2023) and editing methods Geng et al. (2024); Sheynin et al. (2024); Zhang et al. (2024a); Brooks et al. (2023). Specialized techniques via fine-tuning or auxiliary modules now target personalized generation Ruiz et al. (2023); Kumari et al. (2023); Li et al. (2024), text-guided synthesis Chen et al. (2023b; 2024a), example-driven manipulation Yang et al. (2023); Chen et al. (2024b), and human-centric processing Xiao et al. (2023). Recent models like SDXL Podell et al. (2023), ContextDiff Yang et al. (2024c), and DALL-E

3 Betker et al. (2023) enhance text alignment, yet generating photorealistic images from complex prompts—particularly those requiring spatial, attribute, or numerical coherence—remains challenging. We address this by leveraging LLMs to iteratively refine prompts through feedback.

**LLM-based agents and multi-agent cooperation.** LLM-based agents Wang et al. (2024a) autonomously perceive environments, execute actions, and evolve through LLMs' knowledge and reasoning, with systems like AutoGPT Richards et al. (2021) and BabyAGI Nakajima (2023) showcasing advanced decision-making. In T2I generation, multimodal LLM agents Wang et al. (2024b) address complex prompts, while others Qin et al. (2024) employ LLMs for model selection. Multi-agent cooperation enhances domains such as medicine (Medagent Tang et al. (2023) for patient analysis) and software development (MetaGPT Hong et al. (2023) with collaborative roles). Creative collaboration frameworks like AutoAgents Chen et al. (2023a) and AgentVerse Chen et al. (2023d) further demonstrate multi-agent versatility. Despite promising results, current T2I generation research primarily relies on single-agent strategies with limited multi-agent interaction, leaving scope to integrate long-term strategic planning and more coherent cooperation among multiple agents.

**LLMs as prompt optimizers.** LLMs have shown increasing utility as prompt optimizers across various NLP tasks, often employing in-context learning or evolutionary algorithms Zhou et al. (2022); Pryzant et al. (2023); Liu et al. (2023). Over time, prompt optimization approaches have evolved from in-context learning to black-box prompting, which offers greater efficiency by removing the need for internal model access. For example, APE Mañas et al. (2024); Liu et al. (2023) applies LLMs in few-shot language scenarios, while in multimodal settings, LLMs can generate visual descriptors for zero-shot classification Menon & Vondrick (2022). Some work Hu et al. (2023) leverages LLMs to generate VQA queries and bounding box layouts Yang et al. (2024b) to evaluate and improve image quality. Other studies Hao et al. (2024) have leveraged reinforcement learning to enhance image aesthetics and filter out non-visual aspects. In our method, we demonstrate that LLMs can iteratively refine prompts by combining chain-of-thought Wei et al. (2022) reasoning with conversational feedback, effectively serving as optimizers through in-context learning.

## 3 METHOD

We begin by presenting the overall framework of Mac-Tiger for LLM-based T2I agents in Sec. 3.1, where we introduce the need for multi-agent cooperation in handling complex prompts. Due to the limitations of single-agent systems, we propose Mac-Tiger as a solution to optimize the generation process through collaborative roles. In Sec. 3.2, we elaborate on how roles are assigned to each agent to maximize efficiency in the generation process. Finally, in Sec. 3.3, we describe the feedback-driven prompt optimization phase, which allows agents to iteratively refine the prompt and improve the quality of the generated image.

### 3.1 OVERVIEW FRAMEWORK OF MAC-TIGER

Mac-Tiger is designed to optimize the T2I generation process by leveraging the strength of a multi-agent system powered by MLLMs. The primary objective is to create an iterative and reflective process for improving textual prompts which guide an image generation model. This process involves several key agents, notably the *Reviewer*, *Challenger*, and *Refiner*, that function cyclically to refine the text prompt and, subsequently, improve the generated images.

Each agent of Mac-Tiger comprises several modules, as depicted in Fig. 2, including 1) a perception module, 2) a memory module, 3) a communication module, 4) an evaluation module, 5) a cooperative planning module, 6) a feedback adaptation module, and 7) an execution module. The perception module gathers observations from the environment, such as images generated by the T2I model, current and past text prompts, and feedback from other agents. The memory module dynamically stores shared task information, dialogue histories, iterative progress, and revision records in textual format, while also saving all generated images for future reference and evaluation. The communication module extracts relevant information from memory, utilizes an MLLM to generate feedback messages, and transmits these messages to other agents. The evaluation module assesses the generated images against the text prompts and predefined metrics. The feedback adaptation module integrates evaluations and feedback to produce targeted suggestions for prompt optimization. The cooperative planning module consolidates information from memory and the generated images to coordinate

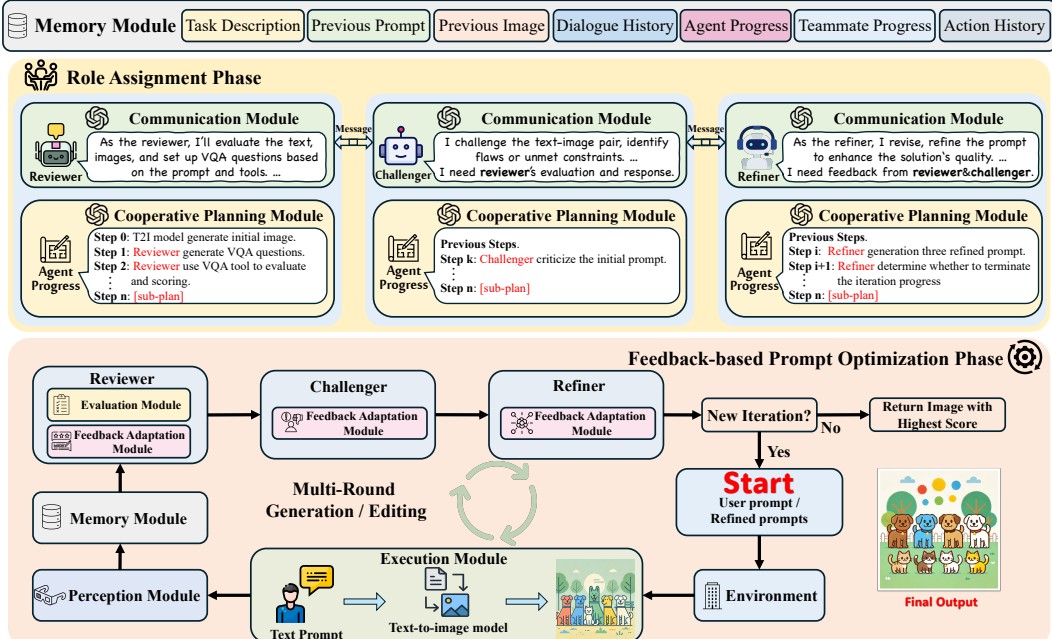

Figure 2: **Overview of the Mac-Tiger framework for T2I multi-agent cooperation.** Mac-Tiger operates in two main phases: 1) **Role Assignment**: The agents—Reviewer, Challenger, and Refiner—collaboratively define their roles before initiating the iterative prompt refinement process. 2) **Feedback-based Prompt Optimization**: Agents cooperatively adopt a feedback adaptation module to iteratively refine the text prompt that improves the generated images over multiple cycles.

feedback and plan workflows, enhancing efficiency and reducing conflicts. Finally, the execution module inputs the optimized prompts into the T2I model to generate images that better align text and image consistency.

To enhance the quality and efficiency of T2I generation tasks, this framework is inspired by human collaborative approaches Tuomela (1998); Thürmer et al. (2017). Utilizing a multi-agent system, it enables iterative optimization of text prompts through the collaborative efforts of agents in distinct roles—*Reviewer*, *Challenger*, and *Refiner*. The workflow consists of two key phases: role assignment and feedback-based prompt optimization. In the role assignment phase, tasks are distributed among agents based on their responsibilities: the *Reviewer* evaluates the quality and consistency of generated images, the *Challenger* identifies overlooked issues critically, and the *Refiner* consolidates feedback to optimize text prompts. This collaborative distribution ensures a solid foundation for iterative refinements. In the feedback-based prompt optimization phase, agents dynamically collaborate to refine text prompts over multiple rounds. Each cycle begins with the *Reviewer* assessing the generated image's alignment with the text prompt and task requirements. The *Challenger* then critiques both the image and the *Reviewer*'s feedback, adding constraints or identifying additional issues. Based on this combined feedback, the *Refiner* adjusts the text prompt to address the identified problems. This iterative process continues until the generated image meets the desired quality standards, such as semantic fidelity and visual coherence. By building on the latest results and collaborative feedback, Mac-Tiger maintains an adaptive and efficient generation process. The following sections provide a detailed explanation of these two phases and illustrate how agents in their respective roles collaborate to drive incremental improvements in generation quality.

## 3.2 ROLE ASSIGNMENT

To generate a long-term plan that coordinates all agents to efficiently accomplish T2I tasks, Mac-Tiger requires a detailed assignment of each agent's roles, including the module allocation within individual agents and the coordination of various modules across multiple agents.

**Reviewer.** As the primary quality evaluator, the *Reviewer* uses the evaluation module to analyze generated images against criteria like prompt alignment and user constraints. The evaluation module utilizes LLM to generate Visual Question Answering (VQA) questions and labels based on the user

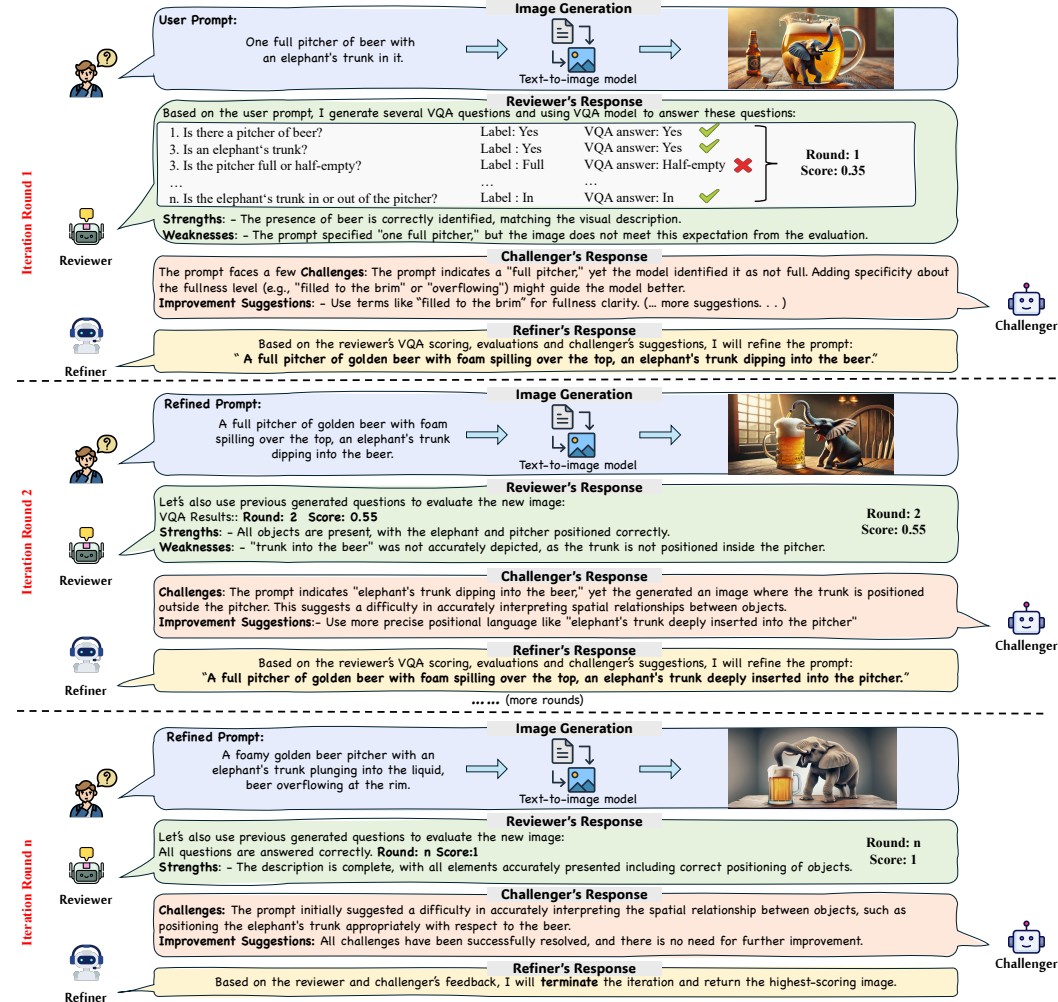

Figure 3: Examples of the evaluation and optimization process of Mac-Tiger via multi-turn discussion.

prompt, employing a VQA model to answer these questions using the generated image. The response accuracy is calculated with the score: $score = \frac{n_c}{N}$, where $n_c$ is the number of correct answers and $N$ the total number of questions, providing a metric for image fidelity and constraints adherence. The perception module aids the *Reviewer* by collecting recent image outputs and contextual signals, storing them in the memory module for continuity across iterations. Through the communication module, the *Reviewer* shares assessments with other agents, notably the *Challenger* and *Refiner*, highlighting necessary areas of attention. Additionally, when coordination on new plans is needed, the *Reviewer* consults the cooperative planning module to align actions with team objectives.

**Challenger.** Positioned as the critical thinker, the *Challenger* uses the results of *Reviewer*'s evaluation module to further probe issues found or possibly overlooked by the *Reviewer*. Its perception module extracts new observations and context, while the memory module preserves not only its own critiques but also any relevant feedback provided by the *Reviewer*. By harnessing the communication module, the *Challenger* engages in dialogues, asking probing questions or highlighting constraints not yet satisfied. These interactions help shape the team's cooperative planning module decisions, enabling adjustments to the overall workflow if previously unconsidered constraints or conflicts arise.

**Refiner.** Acting as the executor of improvements, the *Refiner* merges insights from both the *Reviewer* and the *Challenger* to revise text prompts in a targeted manner. Its feedback adaptation module transforms critiques into tangible modifications to the prompt, addressing areas such as semantic precision or stylistic clarity. The *Refiner* uses the execution module to feed updated prompts back into the T2I model, ultimately generating improved images. During each iteration, the *Refiner*'s perception module and memory module keep track of current outputs and store historical changes,

ensuring consistency throughout the refinement process. In parallel, the cooperative planning module coordinates with other agents to streamline feedback and minimize conflicts or redundant refinements.

Through the complementary collaboration of *Reviewer*, *Challenger*, and *Refiner*, Mac-Tiger leverages each agent's strengths to ensure text prompts undergo rigorous evaluation, critical analysis, and refinement, which progressively produces semantically consistent images through iterations.

### 3.3 FEEDBACK-BASED PROMPT OPTIMIZATION

The iterative feedback loop serves as a high-level guide for refining text prompts, orchestrating the interactions among the *Reviewer*, *Challenger*, and *Refiner* to enhance the quality of generated images. However, as the system processes more images and incorporates new observations, earlier assumptions about style, constraints, or context may become outdated. Significant progress in one round—such as discovering an overlooked detail, resolving contradictory constraints, or aligning the prompt more precisely with user prompts—can render the existing prompt less effective for subsequent iterations. In such cases, the original refinement plan must be revisited to ensure that each agent remains aligned with the latest insights.

To address these dynamically emerging insights, we design a feedback adaptation module that updates the text prompt whenever an agent presents new feedback. This procedure follows a similar cycle of prompt initialization, evaluation, and multi-turn discussion as outlined in earlier sections, but incorporates additional placeholders such as `<Reviewer Feedback>` and `<Challenger Critique>` to capture the evolving perspectives of each agent. Whenever the *Reviewer* or *Challenger* uncovers further ambiguities or conflicting constraints, the *Refiner* promptly revises the prompt, followed by a multi-round discussion that helps the agents reach consensus on how best to incorporate these changes. The prompting scheme for adaptive refinement process include statements such as:

```
Prompt: <Cur_Prompt> + <Cur_Image> + <Reviewer Feedback> + <Challenger
Response> + <Dialog History> \n.  LLM:  <Messages>.
```

In response, the *Refiner* proposes a revised prompt or additional messages for collaborative agents, ensuring that newly uncovered issues are addressed before proceeding to the next iteration.

Once the refined prompt is established through these iterative discussions, each agent proceeds with the updated instructions. This may involve generating a new image, performing another evaluative pass, or consulting the system's memory of previous feedback to ensure consistency. The refinement iteration continues until the *Reviewer*'s evaluation module assigns a score of 1 to an image, or the maximum number of iterations is reached. Upon termination, the process returns image with highest score. Fig. 3 shows the evaluation and optimization process through multi-turn discussions among agents. Detailed prompts for feedback adaptation module are in Appendix. Fig. 9, 10 and 12.

## 4 EXPERIMENTS

To demonstrate the performance of Mac-Tiger, we conduct experiments across two distinct generative scenarios: T2I generation and image editing.

**Benchmarks.** For T2I generation, we primarily perform quantitative comparisons using the recent T2I-CompBench benchmark Huang et al. (2023). This benchmark focuses on generating images from complex text prompts that involve multiple objects, each with its own attributes and relationships. The evaluation covers several key aspects: (i) Attribute Binding, where each text prompt includes multiple attributes that bind to different entities; (ii) Numeric Accuracy, which assesses scenarios where multiple entities share the same class name, with the number of each entity being greater than or equal to two; and (iii) Complex Relationships, which involve multiple entities with various attributes and relationships, including both spatial and non-spatial interactions. For image editing, we primarily use the MagicBrush benchmark Zhang et al. (2024a), which involves multiple types of text instructions for image editing. Following the settings in Zhang et al. (2024a), $L_1$ and $L_2$ are used to measure the average pixel-level absolute difference between the generated image and the ground truth image. CLIP-I and DINO measure image quality through the cosine similarity between the generated image and the reference ground truth image using their CLIP and DINO embeddings.

Table 1: **Quantitative Comparison on T2I-CompBench.** Best scores are highlighted in blue and second-best in green . Mac-Tiger optimization shows significant improvements across all scenarios.

| Model | Attribute Binding | | | Object Relationship | | Complex↑ |
|---|---|---|---|---|---|---|
| | Color↑ | Shape↑ | Texture↑ | Spatial↑ | Non-Spatial↑ | |
| Stable Diffusion v1.4 Rombach et al. (2022) | 0.3765 | 0.3576 | 0.4156 | 0.1246 | 0.3079 | 0.3080 |
| Stable Diffusion v2 Rombach et al. (2022) | 0.5065 | 0.4221 | 0.4922 | 0.1342 | 0.3096 | 0.3386 |
| Composable Diffusion Liu et al. (2022) | 0.4063 | 0.3299 | 0.3645 | 0.0800 | 0.2980 | 0.2898 |
| Structured Diffusion Feng et al. (2022) | 0.4990 | 0.4218 | 0.4900 | 0.1386 | 0.3111 | 0.3355 |
| Attn-Exct v2 Chefer et al. (2023) | 0.6400 | 0.4517 | 0.5963 | 0.1455 | 0.3109 | 0.3401 |
| GORS Huang et al. (2023) | 0.6603 | 0.4785 | 0.6287 | 0.1815 | 0.3193 | 0.3328 |
| DALL-E 2 Ramesh et al. (2022) | 0.5750 | 0.5464 | 0.6374 | 0.1283 | 0.3043 | 0.3696 |
| PixArt-$\alpha$ Chen et al. (2023c) | 0.6886 | 0.5582 | 0.7044 | 0.2082 | 0.3179 | 0.4117 |
| ConPreDiff Yang et al. (2024a) | 0.7019 | 0.5637 | 0.7021 | 0.2362 | 0.3195 | 0.4184 |
| SDXL Podell et al. (2023) | 0.6369 | 0.5408 | 0.5637 | 0.2032 | 0.3110 | 0.4091 |
| Mac-Tiger $_{\text{SDXL}}$ | 0.6831 | 0.5607 | 0.5577 | 0.2299 | 0.3225 | 0.4176 |
| $\Delta$(Margin) | +0.0462 | +0.0199 | -0.0060 | +0.0267 | +0.0115 | +0.0085 |
| DALL-E 3 Betker et al. (2023) | 0.7785 | 0.6205 | 0.7036 | 0.2865 | 0.3003 | 0.3773 |
| Mac-Tiger $_{\text{DALL−E 3}}$ | **0.8017** | **0.6290** | **0.7135** | **0.2877** | **0.3339** | **0.4487** |
| $\Delta$(Margin) | +0.0232 | +0.0085 | +0.0099 | +0.0012 | +0.0336 | +0.0714 |

CLIP-T assesses text-image alignment by examining the cosine similarity between local descriptions and the CLIP embeddings of the generated images.

**Implementations.** In our experiments, GPT-4o accessed through the OpenAI API was employed as the LLM-based agent for the *Reviewer*, *Challenger* and *Refiner* components. The default hyperparameters for the LLMs were conFig.d as follows: a temperature of 0.7, a maximum output token limit of 700, and top-p sampling with $p = 1$. For the *Reviewer* module, mPLUG-large Li et al. (2022) was utilized as the VQA model for performing visual scoring. For T2I generation, we utilized SDXL and DALL-E 3 as the image generation backbones, while for image editing, we selected InstructPix2Pix Brooks et al. (2023) and HIVE Zhang et al. (2024b) as the core image editing frameworks, with a maximum of 3 iterations for termination.

## 4.1 MAIN RESULTS

**Results on text-to-image generation.** Table 1 presents the experimental results of Mac-Tiger on the T2I-CompBench benchmark. We conducted a comparative analysis against state-of-the-art T2I models, including SDXL, DALL-E 3, ConPreDiff, and PixArt-$\alpha$, to validate the effectiveness of the multi-agent collaborative approach in optimizing generation quality. The results demonstrate that Mac-Tiger consistently outperforms existing models across all evaluation scenarios, with significant improvements in handling image details, complex relationships, and text-image alignment. Traditional T2I models like SDXL and DALL-E 3 show limitations in numerical constraints and complex relationships, while our multi-agent framework progressively refines outputs through the *Reviewer*'s meticulous reviews, *Challenger*'s in-depth challenges, and *Refiner*'s prompt optimizations. The iterative feedback significantly enhances image quality, with the *Refiner*'s updates proving critical for color accuracy, spatial coherence, and attribute matching. Furthermore, the text-to-image consistency of models like SDXL and DALL-E 3 was markedly enhanced through our framework, showing improved accuracy in attribute binding and complex relationships.

Table 2: **Quantitative Comparison of Text-Guided Image Editing Models on the MagicBrush Benchmark.** We denote the best score in blue , and the second-best score in green .

| Methods | L1↓ | L2↓ | CLIP-I↑ | DINO↑ | CLIP-T↑ |
|---|---|---|---|---|---|
| Open-Edit | 0.1430 | 0.0431 | 0.8381 | 0.7632 | 0.2610 |
| SD-SDEdit | 0.1014 | 0.0278 | 0.8526 | 0.7726 | 0.2777 |
| InstructPix2Pix | 0.1122 | 0.0371 | 0.8524 | 0.7428 | 0.2764 |
| w/ Mac-Tiger | **0.0975** | 0.0321 | **0.8587** | 0.7471 | 0.2783 |
| $\Delta$(Margin) | +0.0147 | +0.0050 | +0.0063 | +0.0043 | +0.0019 |
| HIVE | 0.1092 | 0.0341 | 0.8519 | 0.7500 | 0.2752 |
| w/ Mac-Tiger | 0.1010 | **0.0276** | 0.8561 | **0.7737** | **0.2799** |
| $\Delta$(Margin) | +0.0082 | +0.0065 | +0.0042 | +0.0237 | +0.0047 |

**Results on text-guided image editing.** In Table 2, we compare the performance of Mac-Tiger with state-of-the-art image editing models, Open-Edit Liu et al. (2020), SD-SDEdit Meng et al. (2021), InstructPix2Pix, and HIVE. The results demonstrate that Mac-Tiger excels in handling complex image editing tasks. Unlike traditional approaches, Mac-Tiger employs three LLM-based agents—*Reviewer*, *Challenger*, and *Refiner*—that collaboratively address local deficiencies through multi-modal reasoning and iterative refinement. These agents dynamically exchange feedback, identify ambiguities, and optimize editing instructions by treating images as contextual inputs. This complementary interaction enhances both local detail modifications and global consistency, achieving superior detail handling and instruction optimization in complex scenarios.

## 4.2 ABLATION STUDY

Table 3: Ablation study of T2I Generation on the T2I-CompBench. Best score in blue .

| Model | Attribute Binding | | | Object Relationship | | Complex↑ |
|---|---|---|---|---|---|---|
| | Color↑ | Shape↑ | Texture↑ | Spatial↑ | Non-Spatial↑ | |
| DALL-E 3 | 0.7785 | 0.6205 | 0.7036 | 0.2865 | 0.3003 | 0.3773 |
| w/o *Reviewer* | 0.7829 | 0.6199 | 0.7047 | 0.2834 | 0.3029 | 0.3829 |
| w/o *Challenger* | 0.7995 | 0.6259 | 0.7102 | **0.2879** | 0.3237 | 0.4249 |
| Mac-Tiger | **0.8017** | **0.6290** | **0.7135** | 0.2877 | **0.3339** | **0.4487** |

**Impact of the *Reviewer*.** We utilizes DALL-E 3 as the T2I model for ablation study. As shown in Table 3, the absence of the *Reviewer* (w/o *Reviewer*) leads to a significant drop in performance across all scenarios. The *Reviewer* plays a critical role in evaluating the generated images, identifying local deficiencies, and providing targeted feedback for improvements. Without the *Reviewer*, the optimization process lacks precise and high-quality assessments, resulting in suboptimal editing instructions that fail to adequately address the flaws in the images. This highlights the *Reviewer*'s essential contribution to ensuring high-quality outputs.

**Impact of the *Challenger*.** Similarly, as shown in Table 3, removing the *Challenger* (w/o *Challenger*) results in a moderate decline in performance. The *Challenger* primarily enhances the optimization process by questioning and identifying potential problems, ensuring broader coverage of issues and potential areas for improvement. In the absence of the *Challenger*, the model can still rely on the *Reviewer* and *Refiner* to maintain a certain level of optimization. However, some deeper issues or opportunities for refinement may be overlooked. Thus, while the impact of the *Challenger* is less pronounced, it plays an important role in improving the comprehensiveness and robustness of the framework.

**Impact of iterative rounds.** Fig. illustrates the effect of the number of iterations on the performance of iterative image editing. Experiments were conducted using the complex and color datasets of T2I-CompBench. By leveraging multiple rounds of iterative feedback and refinement, Mac-Tiger achieves significant improvements in editing performance. 4 This enhancement demonstrates that Mac-Tiger effectively integrates historical multimodal context into the optimization process. However, due to the limitations of LLM-based agent

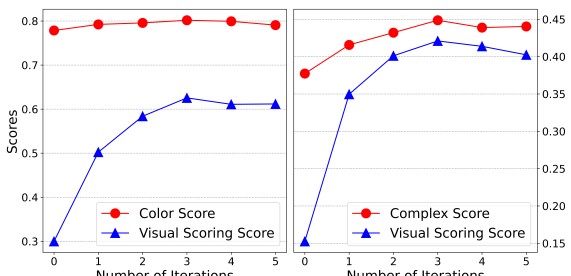

Figure 4: **Performance of Mac-Tiger across iterative rounds on T2I-CompBench.** Left is color dataset, and right is complex dataset. The optimal performance is achieved at iteration 3.

memory and reasoning capabilities, as well as the coordination inefficiencies introduced by multi-agent collaboration, increasing the number of iterations does not always result in better outcomes. Performance peaks at the third iteration and gradually declines in subsequent iterations. This finding underscores the importance of balance in iterative optimization strategies, as excessive iterations may lead to reduced efficiency or overfitting, ultimately compromising overall editing quality.

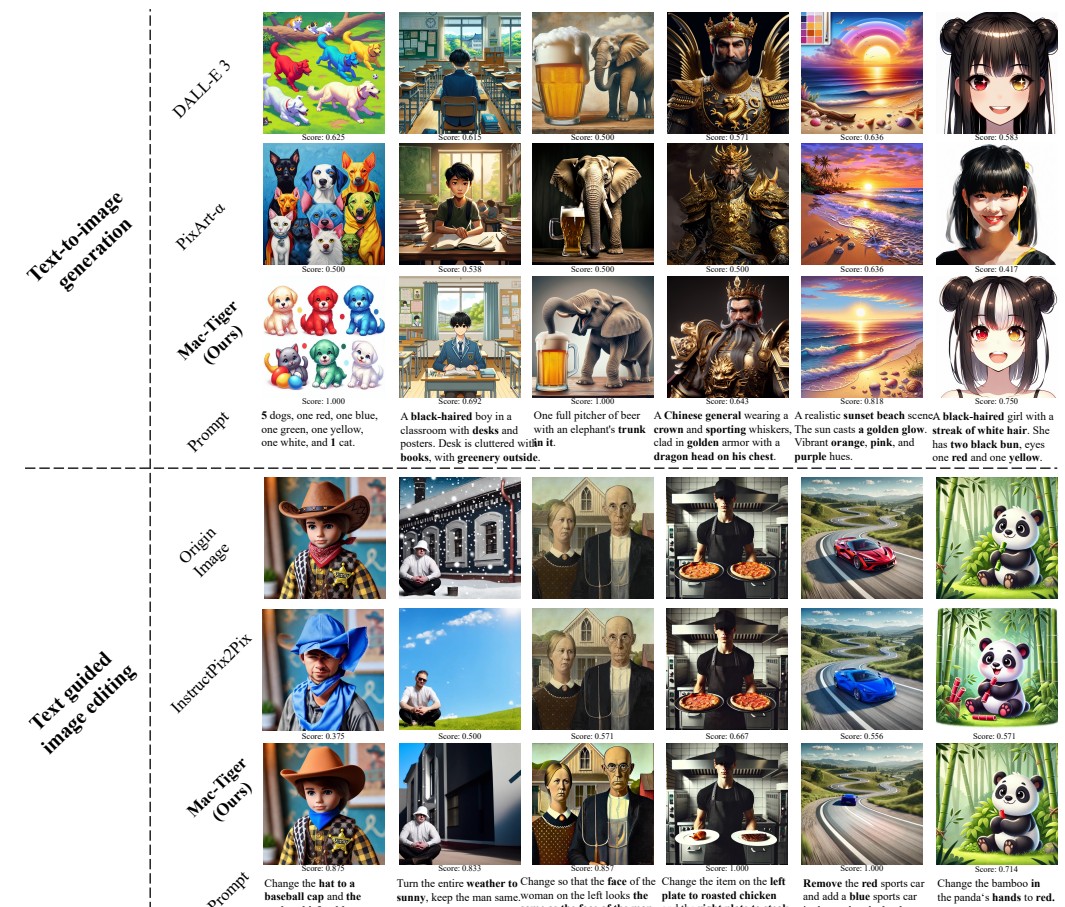

Figure 5: **Qualitative comparison with existing SOTA methods for T2I generation and image editing.** The last row represents the user instruction prompt. Mac-Tiger demonstrates significantly better consistency and accuracy with the origin image and user instruction.

**Comparative visualized results.** Fig. 5 demonstrate Mac-Tiger' superiority over SOTA models in T2I generation and image editing. In T2I tasks, Mac-Tiger excels in handling complex spatial relationships (e.g., the third example "`beer with an elephant's trunk in it`"), where PixArt-$\alpha$, SDXL, and DALL-E 3 fail to resolve spatial intricacies, while our multi-agent framework iteratively optimizes prompts via visual scoring to achieve coherent outputs. For image editing, Mac-Tiger precisely binds attributes and preserves contextual integrity (e.g., keeping "`the man`" unchanged in the second example), outperforming existing methods in target-specific modifications. Spatial accuracy is further enhanced through visual error detection and multimodal in-context learning, as shown in positioning tasks (third/fifth examples). These results validate Mac-Tiger' robustness in both generation and editing through iterative refinement and multimodal reasoning. More case studies and visualisations are detailed in Appendix A.

## 5 CONCLUSION

In this paper, we propose Mac-Tiger, a multi-agent framework that synergistically combats limitations of single-agent methods in text-to-image generation through iterative collaboration among Reviewer, Challenger, and Refiner agents. The closed-loop workflow unifies prompt evaluation and refinement, demonstrating superior task completion accuracy and image quality on T2I-CompBench and MagicBrush, while maintaining compatibility with existing diffusion architectures. Future work will extend the framework for multi-modal inputs (e.g., layout/semantic guidance) and optimize real-time efficiency for industrial design applications.

**Ethics statement.** We confirm that this work aligns with accepted ethical standards in machine learning research. All data and methodologies used are publicly available or properly cited.

**Reproducibility statement.** To support reproducibility, we have provided full details of our experimental setup, including hyperparameters and dataset descriptions, in the experimental section. Code is available.

**The use of large language models (LLMs).** We utilize LLMs to assist and enhance our writing. They help us improve the quality and effectiveness of our textual expression.

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

## A  MORE EXPERIMENTS

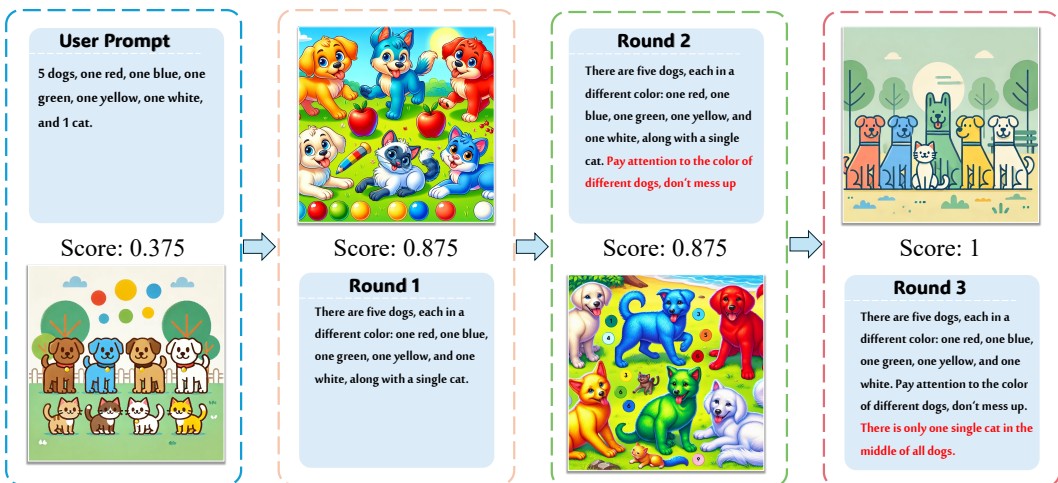

Figure 6: Visualization of Mac-Tiger refinement iteration progress in T2I generation tasks.

**Case study of Mac-Tiger refinement.**   To illustrate the effectiveness of our proposed method, we present a text-to-image (T2I) generation example in Fig. 6. In this case, the *Reviewer* first generates QA-pairs to evaluate the consistency between text and image based on the user prompt, demonstrated below. Given that the prompt includes various colors and quantities, it presents a significant challenge for the T2I model (DALL-E 3). Initially, the T2I model generates an image with incorrect colors and quantities of dogs and cats, resulting in a low Visual Scoring score, indicating poor text-image consistency. Throughout each iteration of optimization, MLLM utilizes in-context learning and chain of thought techniques to refine the prompt by correcting insufficient representations and highlighting key aspects such as quantity (`"only one single cat"`) and color (`"Pay attention to the color"`). With each iterative step, the Visual Scoring score improves, leading to enhanced text-image consistency.

**Visualized results about image editing tasks with complex user instructions.**   To demonstrate that our Mac-Tiger is capable of recognizing and following complex user instructions in image editing tasks, we provide visual examples in Fig. 8. Leveraging multi-round Visual Scoring and multimodal in-context learning, our framework effectively addresses complex requirements that are challenging to achieve in a single generation. For instance, Mac-Tiger is capable of identifying and refining specific areas of an image that require modification through multiple rounds of adjustments, ensuring that other elements remain unchanged (see Fig. 8 top left and bottom). This precision is facilitated by the global image scoring mechanism inherent in Visual Scoring. Additionally, Mac-Tiger can adjust the overall style and tone of the image while also understanding user instructions that require reasoning abilities (see Fig. 8 top right), which further underscores the critical role of multi-round Dynamic Prompt Optimization. Finally, Mac-Tiger excels in fine-grained tasks, such as precise modification and transformation of text within an image, including text content, font, color, and layout (see Fig. 8 center). These visualization examples clearly demonstrate the efficacy of our approach in achieving precise and targeted image editing. Fig. 7 also shows that Mac-Tiger exhibits superior capabilities in generating fine-grained details.

Here are questions generated by the *Reviewer*:

```
Element 1: dogs
Element_type: animal
Question: Are there dogs in the image?
Choices: ['yes', 'no']
Answer: yes

Element 2: dogs
Element_type: animal
Question: How many dogs are present in the image?
```

```
Choices: ['1', '3', '5', '7']
Answer: 5

Element 3: cat
Element_type: animal
Question: How many cats are in the image?
Choices: ['0', '1', '2', '3']
Answer: 1

Element 4: red
Element_type: color
Question: Is one of the dogs red?
Choices: ['yes', 'no']
Answer: yes

Element 5: blue
Element_type: color
Question: Is one of the dogs blue?
Choices: ['yes', 'no']
Answer: yes

Element 6: green
Element_type: color
Question: Is one of the dogs green?
Choices: ['yes', 'no']
Answer: yes

Element 7: yellow
Element_type: color
Question: Is one of the dogs yellow?
Choices: ['yes', 'no']
Answer: yes

Element 8: white
Element_type: color
Question: Is one of the dogs white?
Choices: ['yes', 'no']
Answer: yes
```

Generate image with Mac-Tiger      Image Editing with Mac-Tiger

Figure 7: **Samples generated by Mac-Tiger** show outstanding quality, showcasing impressive fidelity and accuracy.

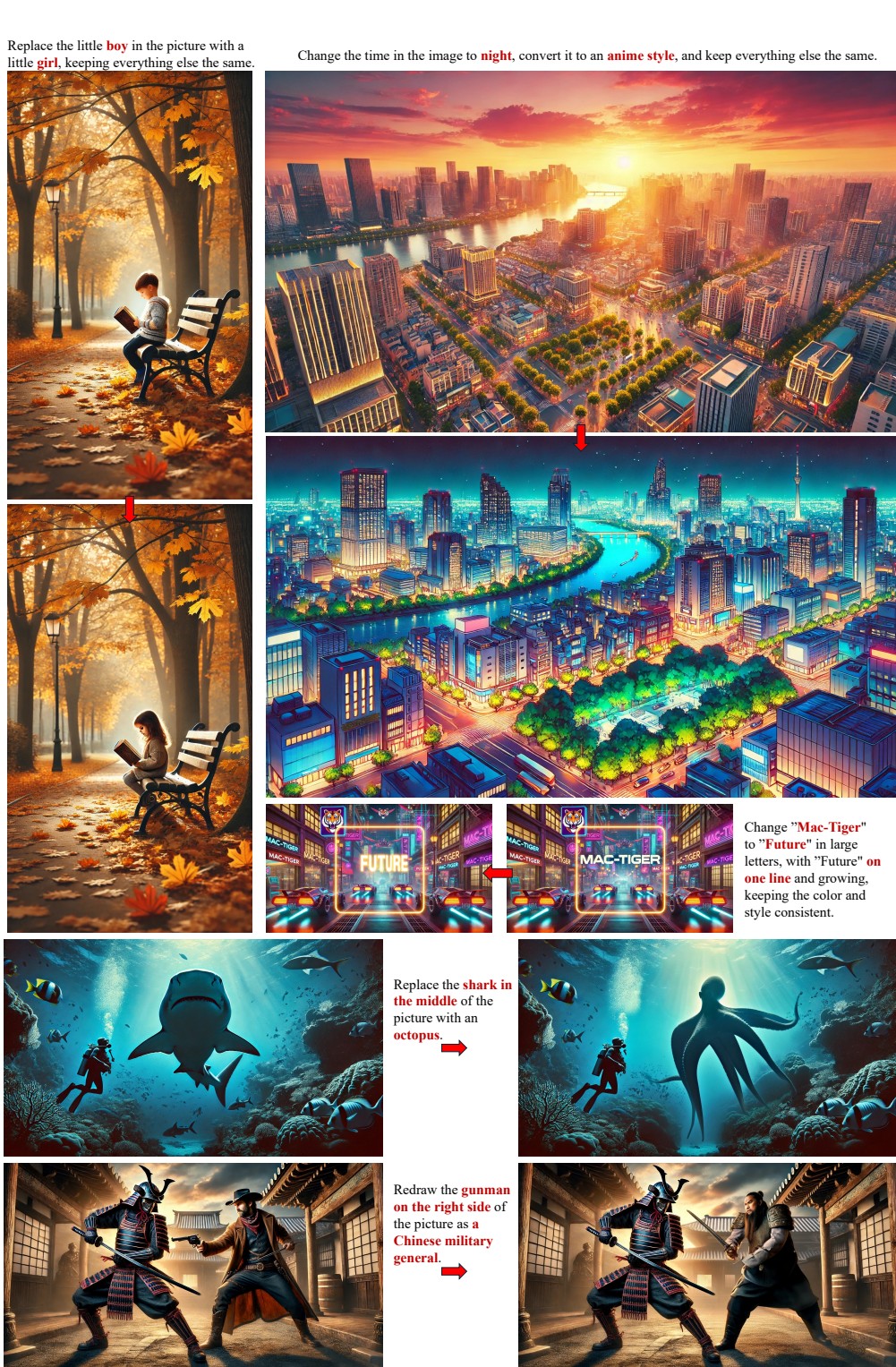

Figure 8: Visualized results of Mac-Tiger for image editing with complex user instructions.

# B  PROMPT TEMPLATE

We list the prompts template for *Reviewer*, *Challenger*, *Refiner*, and prompt for generating questions in the evaluation module as follows.

---

You are a prompt reviewer for text-to-image models. Your role is to evaluate both the initial human-written prompt and previous prompts based on their effectiveness in conveying visual elements that match the generated images. Consider the scores assigned to each visual element in the outputs, with 1 indicating a perfect match and 0 indicating no match.

Your task is to review the initial prompt: "{user_prompt}". Additionally, provide an evaluation of the previous prompts given.

Here is the image that the text-to-image model generated based on the initial prompt:
{{image_placeholder}}

Here are the previous prompts and their visual element scores:
## Previous Prompts
{previous_prompts}
## Visual Element Scores
{VS_results}

Provide a comprehensive evaluation of the initial prompt and each of the previous prompts. Focus on the correctness and completeness of each prompt in relation to the generated images, highlighting strengths and weaknesses. Depth questioning or suggested alterations are not necessary, but insightful commentary is encouraged.

If there are no previous prompts, simply provide an evaluation for the initial prompt. Respond with each evaluation in between <EVALUATION> and </EVALUATION> as follows:
1. <EVALUATION>Your Evaluation for initial prompt</EVALUATION>
2. <EVALUATION>Your Evaluation for previous prompt 1</EVALUATION>
...
n. <EVALUATION>Your Evaluation for previous prompt n</EVALUATION>

---

Figure 9: Prompts for Agent *Reviewer*.

---

You are a prompt challenger for text-to-image models. Your role is to critically evaluate the initial human-written prompt and previous prompts, identifying potential flaws and constraints that are not met based on the evaluation of the reviewer. Consider the scores assigned to each visual element in the outputs, with 1 indicating a perfect match and 0 indicating no match.

Your task is to challenge the initial prompt: "{user_prompt}". Additionally, provide a critique of the previous prompts.

Here is the image that the text-to-image model generated based on the initial prompt:
{{image_placeholder}}

Here are the previous prompts and their visual element scores:
## Previous Prompts
{previous_prompts}
## Visual Element Scores
{VS_results}
## Reviewer's Evaluation
{reviewer_evaluation}

Based on the correctness and completeness of each prompt in relation to the generated images, identify potential weaknesses and unmet constraints. Propose improvement ideas or introduce new counterexamples and constraints to test the current solutions.
If there are no previous prompts, focus on challenging the initial prompt. Respond with each challenge in between <CHALLENGE> and </CHALLENGE> as follows:
1. <CHALLENGE>Your Challenge for initial prompt</CHALLENGE>
2. <CHALLENGE>Your Challenge for previous prompt 1</CHALLENGE>
...
n. <CHALLENGE>Your Challenge for previous prompt n</CHALLENGE>

---

Figure 10: Prompts for Agent *Challenger*.

You are a prompt refiner for text-to-image models. Your role is to improve the quality of the initial human-written prompt and previous prompts by incorporating feedback received from the reviewer and challenger. Your goal is to adjust, refine, and reconstruct the prompts to better meet the intended requirements and constraints.

Your task is to refine the initial prompt: "{user_prompt}" and the previous prompts based on the feedback received.

Here is the image that the text-to-image model generated based on the initial prompt:
{{image_placeholder}}

Here are the previous prompts and their visual element scores:
## Previous Prompts
{previous_prompts}
## Visual Element Scores
{VS_results}
## Reviewer's Evaluation
{reviewer_evaluation}
## Challenger's Challenge
{challenger_response}

Using the feedback from both the reviewer and the challenger, modify and enhance the prompts to address weaknesses and fulfill unmet constraints. Generate improved prompts that capture the intended visual elements more effectively. If there are no previous prompts, focus on refining the initial prompt. Respond with each refined prompt in between <REFINED_PROMPT> and </REFINED_PROMPT> as follows:

<REFINED_PROMPT>Your Refined prompt</REFINED_PROMPT>

Figure 11: Prompts for Agent *Refiner*.

Given a image descriptions, generate one or two multiple-choice questions that verifies if the image description is correct.
Classify each concept into a type (object, human, animal, activity, counting, color, spatial, other), and then generate a question for each type.

###
Description: A cat playing with a blue ball on a wooden floor next to a table.
- Entities: cat, ball, floor, table
- Activities: playing
- Colors: blue
- Counting:
- Other attributes: wooden

Questions and answers are below:

About cat (animal):
Q: Is there a cat?
- Choices: yes, no
- A: yes

About ball (object):
Q: Is the ball blue?
- Choices: yes, no
- A: yes
Q: What is the cat playing with?
- Choices: ball, string, toy, mouse
- A: ball

About floor (spatial):
Q: Is the cat on a wooden floor?
- Choices: yes, no
-  A: yes

About playing (activity):
Q: Is the cat playing?
- Choices: yes, no
- A: yes

###
Description: A man in a red shirt is walking with three dogs in the park.
- Entities: man, shirt, dogs, park
- Activities: walking
- Colors: red
- Counting: three
- Other attributes:

Questions and answers are below:

About man (human):
Q: Is there a man?
- Choices: yes, no
- A: yes

About shirt (object):
Q: What color is the man's shirt?
- Choices: red, blue, green, yellow
- A: red

About dogs (animal):
Q: Are there three dogs?
- Choices: yes, no
- A: yes
Q: How many dogs are there?
- Choices: 1, 2, 3, 4
- A: 3

About walking (activity):
Q: Is the man walking with the dogs?
- Choices: yes, no
- A: yes

About park (spatial):
Q: Is the man walking in the park?
- Choices: yes, no
- A: yes

###
Description: A woman holding a green umbrella is standing near a tree in a rainy street.
- Entities: woman, umbrella, tree, street
- Activities: holding, standing
- Colors: green
- Counting:
- Other attributes: rainy

Questions and answers are below:

About woman (human):
Q: Is there a woman?
- Choices: yes, no
- A: yes

About umbrella (object):
Q: Is the umbrella green?
- Choices: yes, no
- A: yes
Q: What is the woman holding?
- Choices: umbrella, bag, book, phone
- A: umbrella

About tree (object):
Q: Is there a tree near the woman?
- Choices: yes, no
- A: yes

About standing (activity):
Q: Is the woman standing?
- Choices: yes, no
- A: yes

###
Description:

Figure 12: Prompt for generating questions in the evaluation module.

## C  LIMITATIONS

While Mac-Tiger demonstrates strong performance in both text-to-image generation and image editing, it relies heavily on LLM-based agents, which can incur substantial computational overhead and latency during multi-round interactions. Furthermore, prompt refinement effectiveness may degrade with increasing iteration rounds due to memory limitations and potential feedback saturation. In future work, we plan to (i) explore lightweight agent models for real-time deployment, (ii) integrate additional modalities such as layout and depth cues to guide generation more explicitly, and (iii) develop hierarchical agent structures to scale the framework for long-horizon multi-turn creative tasks.

## D  BROADER IMPACT

This paper presents work whose goal is to advance the field of Machine Learning. There are many potential societal consequences of our work, none which we feel must be specifically highlighted here.

