# OpenReview forum: "Mac-Tiger: Multi-Agent Cooperation for Enhanced Text-to-Image Generation"
_ICLR.cc/2026/Conference — Submitted to ICLR 2026_

### Official Review · Reviewer_Gzjq · 2025-10-31

**Soundness:** 2
**Presentation:** 2
**Contribution:** 2
**Rating:** 2
**Confidence:** 4

**Summary:**

The paper proposes Mac-Tiger for complex compositional T2I generation task. Mac-Tiger employs a tri-agent system, including Reviewer,
Challenger, and Refiner roles, that evaluates and refines prompts via dynamic feedback loops and multimodal analysis. The agents first generate initial solutions by exploring diverse perspectives and evaluating other's outputs. Then, the agents iteratively refine the prompt by addressing gaps and inconsistencies identified during the review and challenge phases. Experiments show effectiveness on T2I-CompBench and MagicBrush.

**Strengths:**

1. The paper is well-written and easy to follow.
2. Experiments are extensive for two benchmarks (T2I-CompBench, MagicBrush).

**Weaknesses:**

1. My major concern is about the lack of novelty: There are many other multi-agent works for T2I/T2V generation that share similar ideas (GenArtist, MovieAgent, MAViS).
[A] Wang Z, Li A, Li Z, et al. Genartist: Multimodal llm as an agent for unified image generation and editing[J]. Advances in Neural Information Processing Systems, 2024, 37: 128374-128395.
[B] Wu W, Zhu Z, Shou M Z. Automated movie generation via multi-agent cot planning[J]. arXiv preprint arXiv:2503.07314, 2025.
[C] Wang Q, Huang Z, Jia R, et al. MAViS: A Multi-Agent Framework for Long-Sequence Video Storytelling[J]. arXiv preprint arXiv:2508.08487, 2025.

2. The method relies heavily on the SoTA LLM, which will not generalize to less capable LLMs.

3. The refinement process will introduce additional time cost.

**Questions:**

1. What are the differences among the multi-agent works (GenArtist, MovieAgent, MAViS)?
2. What are the performances of open-source models?
3. What are the time/money costs for each agent?

---

### Official Review · Reviewer_qT1F · 2025-11-01

**Soundness:** 3
**Presentation:** 2
**Contribution:** 2
**Rating:** 4
**Confidence:** 4

**Summary:**

This paper introduces MAC-TIGER, a multi-agent prompt refinement framework for text-to-image (T2I) generation and editing. The system consists of three roles: Reviewer, Challenger, and Refiner, which iteratively critique and improve prompts before sending them to a diffusion model. The method is training-free and model-agnostic. Experiments on T2I-CompBench and MagicBrush show improved compositional correctness and editing fidelity over standard prompting and simple self-refinement loops. Ablations demonstrate the importance of each agent's role and iteration count.

**Strengths:**

Clear and modular tri-agent design that maps to real T2I failure patterns (missing entities, incorrect relations, vague prompts).

Training-free and model-agnostic pipeline, compatible with various diffusion and editing models.

Demonstrated improvements on compositional benchmarks and instruction-guided editing tasks.

**Weaknesses:**

The paper does not contrast well with other multi-agent T2I frameworks. I'm not clear what the main contribution of this paper compared to them.

Missing comparison to recent multi-agent T2I systems, which substantially weakens novelty claims. Relevant works include:

1. GenArtist (NeurIPS 2024) - MLLM-driven multi-stage generation & editing

2. Proactive Agents (ICML 2025) - interactive multi-turn T2I refinement

3. Anywhere (AAAI 2025) - multi-agent foreground-conditioned generation

4. MCCD (CVPR 2025) - multi-agent compositional diffusion

Limited discussion of computational overhead, prompt token budget, inference latency, and API cost.

Mostly automatic metrics; lacks rigorous human preference study.

**Questions:**

Please add a direct comparison with other multi-agent T2I baselines (e.g., GenArtist or MCCD).

Provide scaling analysis: performance vs. number of iterations and prompt length.

Report cost/latency per iteration and total token usage.

---

### Official Review · Reviewer_YkWB · 2025-11-01

**Soundness:** 2
**Presentation:** 3
**Contribution:** 2
**Rating:** 2
**Confidence:** 4

**Summary:**

This paper introduces a multi-agent framework for text-to-image generation. Particularly, it designs three agents (based on GPT-4o) that refine the generated results iteratively: Given an initial synthesized image, a Reviewer is used to rate the image. After that, a Challenger receives the feedback from the Reviewer and finds unsatisfied objectives. Subsequently, the outputs of the reviewer and the challenger are fed into a refiner to generate more specific prompts. The proposed method is evaluated through T2I generation and editing on two public benchmarks, which improves the performance of SDXL and DALL-E 3.

**Strengths:**

1. Three agents are designed to cooperate and facilitate image editing. All of them employ off-the-shelf models directly; hence, the proposed method can be implemented easily.

2. Curated prompts are designed to analyze and refine user instructions.

3. The proposed method supports both T2I generation and editing.

**Weaknesses:**

1. The technical contributions of the proposed framework are limited. Essentially, there is no learning algorithm or optimizable module in the proposed framework to ensure that the final results can be improved. In fact, mainstream methods for this topic are exploring reinforcement learning (like GRPO) and unified models for multimodal understanding and generation.

2. From the quantitative comparison and the ablation study, the performance gain of the proposed method is marginal.

3. The proposed method relies on multiple large models (e.g., GPT-4o), causing substantial computational cost.

**Questions:**

1. How does the base model of the agents affect the performance? Has any open-source model (e.g., QwenVL) been tested?

---

### Official Review · Reviewer_6QU6 · 2025-11-01

**Soundness:** 3
**Presentation:** 3
**Contribution:** 2
**Rating:** 4
**Confidence:** 3

**Summary:**

This paper introduces Mac-Tiger, a multi-agent cooperation framework designed to enhance text-to-image (T2I) generation, particularly for complex prompts that challenge existing models with attribute binding, spatial relationships, and numerical precision. The framework employs a tri-agent system—comprising a Reviewer, a Challenger, and a Refiner—powered by Multimodal Large Language Models (MLLMs). These agents collaborate through iterative feedback loops to evaluate and optimize text prompts, thereby improving the final generated image's quality and alignment with the initial request. The authors demonstrate the framework's effectiveness by applying it to state-of-the-art models like SDXL and DALL-E 3 and evaluating its performance on the T2I-CompBench and MagicBrush benchmarks.

**Strengths:**

1. Well-Defined Problem: The paper addresses a significant and well-recognized limitation in the T2I field: the difficulty of current models in handling compositional complexity. The motivation is clear and targets a critical area for advancement.

2. Novel Conceptual Framework: The application of a structured, multi-agent system to the problem of T2I prompt optimization is a novel idea. The division of labor among the Reviewer, Challenger, and Refiner roles provides a systematic and interpretable approach to iterative prompt refinement.

3. Strong Empirical Results: The quantitative results are a key strength. The framework consistently improves the performance of powerful foundation models (SDXL, DALL-E 3) across all categories of the challenging T2I-CompBench benchmark. This demonstrates the practical effectiveness of the proposed method.

4. Methodological Generality: The framework is shown to be effective not only for T2I generation but also for text-guided image editing, improving the performance of models like InstructPix2Pix and HIVE on the MagicBrush benchmark. This suggests the approach is versatile and not limited to a single task.

**Weaknesses:**

1. While the framework is novel, its core is essentially a sophisticated prompt engineering pipeline that orchestrates existing, off-the-shelf models (GPT-4V as the agent brain, mPLUG-large as the VQA tool). The paper does not address the deeper technical challenges of how to guarantee diverse, non-redundant, and insightful feedback from the agents. The justification for a three-agent system over a simpler two-agent (e.g., critic-refiner) or single-agent self-correction loop is not rigorously established. The ablation study shows each component is helpful but does not prove this specific architecture is optimal.

2. The entire iterative optimization process is guided by a quantitative score from the Reviewer's VQA module. However, this introduces a major point of failure. VQA models themselves are known to be fallible, especially on the same complex compositional queries (e.g., counting, nuanced spatial relations) that the T2I system is trying to solve. The paper treats the VQA score as ground truth without providing any analysis of the VQA model's own accuracy on this task or discussing how the system would handle incorrect feedback. This is a critical technical oversight.

3. Figure 4 shows that performance peaks at 3 iterations and then begins to decline. The authors' explanation—blaming LLM limitations and coordination inefficiencies—is superficial and insufficient. This phenomenon points to a core technical problem, possibly related to prompt overfitting or the feedback loop becoming counter-productive. The paper fails to analyze the root cause of this degradation or propose a mechanism to mitigate it, which is a significant weakness.

4. Unfair Comparison and Lack of Cost Analysis: The experiments compare a baseline model's single inference pass (e.g., "SDXL") with the full Mac-Tiger system ("Mac-Tiger + SDXL"), which involves multiple rounds of expensive GPT-4V calls, VQA model inferences, and T2I generations. This is not a fair comparison. The paper completely omits any discussion of the substantial overhead in terms of latency, computational cost, and API fees. This makes it impossible to assess the method's practical utility. A more rigorous baseline would involve a single-agent iterative method with a comparable computational budget.

5. For a generative task where the output is visual and subjective, relying exclusively on automated metrics is insufficient. A critical missing piece is a human evaluation study. Do human evaluators agree that the images produced by Mac-Tiger are of higher quality and better follow the prompt's instructions compared to the baseline? Without this, the claims of "higher-quality" and "more context-aware" generation are not fully substantiated.

6. The ablation study is too limited. It confirms that removing the Reviewer or Challenger hurts performance but fails to answer more critical questions. An ablation comparing the tri-agent setup to a dual-agent (combining Reviewer and Challenger) or a single-agent self-critique loop is necessary to justify the architectural complexity. Additionally, the sensitivity of the system to the choice of the VQA model or the specific prompting templates used for the agents is not explored.

**Questions:**

1. Could you provide a deeper analysis of why performance declines after three iterations? Have you analyzed the prompts from later iterations to identify if they become overly constrained, verbose, or contradictory? Is there a principled mechanism to automatically terminate the process to avoid this "over-optimization"?

2. How do you address the fact that your core evaluation signal from the mPLUG-large VQA model may be inaccurate? Have you measured the VQA model's accuracy on the T2I-CompBench evaluation set itself? How does the system ensure it is not optimizing towards an incorrect objective set by a fallible VQA model?

3. Can you provide stronger evidence to justify the necessity of having separate Reviewer and Challenger agents, as opposed to a single, more capable "Evaluator-Critic" agent? What specific failure modes does the Challenger address that the Reviewer consistently misses?

4. Can you quantify the computational overhead of your framework? For a typical run of 3 iterations, what is the increase in cost (in terms of LLM/VQA API calls, GPU hours, and wall-clock time) compared to a single inference from the baseline T2I model?

5. Have you considered conducting a human preference study to validate your results? How can you be certain that the improvements in automated metrics translate into a meaningfully better experience and output quality for a human user?

---

### Meta-Review · Area_Chair_YCob · 2026-01-03

**Summary:**

This paper introduces a multi-agent prompt refinement framework for text-to-image generation tasks.

The reviewers shared their concerns on limited novelty/technical contributions, missing related work, unfair baselines/cost analyses, reliability, lack of human studies and weak ablation studies.

The authors did not provide a rebuttal.

I recommend rejection.

**Reviewer Concerns:**

The authors did not provide a rebuttal so all concerns are still outstanding.

**Reviewer Scores:**

The authors did not provide a rebuttal. The reviewers would not have changed their scores.

---

### Decision · Program_Chairs · 2026-01-26

Reject